# Self-Supervised Dam Deformation Anomaly Detection Based on Temporal–Spatial Contrast Learning

**DOI:** 10.3390/s24175858

**Published:** 2024-09-09

**Authors:** Yu Wang, Guohua Liu

**Affiliations:** College of Civil Engineering and Architecture, Zhejiang University, Hangzhou 310058, China; wangyu97@zju.edu.cn

**Keywords:** dam health monitoring, dam deformation, anomaly detection, self-supervised learning

## Abstract

The detection of anomalies in dam deformation is paramount for evaluating structural integrity and facilitating early warnings, representing a critical aspect of dam health monitoring (DHM). Conventional data-driven methods for dam anomaly detection depend extensively on historical data; however, obtaining annotated data is both expensive and labor-intensive. Consequently, methodologies that leverage unlabeled or semi-labeled data are increasingly gaining popularity. This paper introduces a spatiotemporal contrastive learning pretraining (STCLP) strategy designed to extract discriminative features from unlabeled datasets of dam deformation. STCLP innovatively combines spatial contrastive learning based on temporal contrastive learning to capture representations embodying both spatial and temporal characteristics. Building upon this, a novel anomaly detection method for dam deformation utilizing STCLP is proposed. This method transfers pretrained parameters to targeted downstream classification tasks and leverages prior knowledge for enhanced fine-tuning. For validation, an arch dam serves as the case study. The results reveal that the proposed method demonstrates excellent performance, surpassing other benchmark models.

## 1. Introduction

Structural health monitoring (SHM) is essential for evaluating the safety and operational status of buildings, playing a key role in preserving structural integrity and ensuring the safety of human lives [1,2]. DHM serves as a specialized application of SHM, addressing the challenges associated with the aging and performance of dam structures. The importance of dam safety has grown in light of numerous accidents and failures, underscoring the need for effective health monitoring and early warning systems. Dam failure represents a dynamic, time-evolving process, necessitating continuous surveillance to ensure structural integrity [3]. Throughout the operational life, dams are subjected to the dual effects of external loads and the gradual aging of internal materials [4], leading to a gradual decline in structural safety [5]. This degradation typically manifests as anomalies or sudden changes in monitoring data, with deformation being a critical metric for assessing a dam’s operational health [6]. Therefore, conducting in-depth studies on the patterns of dam deformation and developing robust anomaly detection models are crucial for evaluating structural safety and implementing timely interventions.

Dam monitoring systems, which include pendulums installed within and on the dam’s surface, are pivotal for collecting deformation data [7]. However, these data often include outliers that deviate from expected patterns due to complex factors like physical laws, structural defects, and instrumentation errors, complicating accurate outlier identification. These outliers can be categorized as either reasonable or unreasonable based on their origins. Like other monitoring variables, the deformation series exhibits several characteristics, including a large number of monitoring points, rich information content, and long-term and cyclical changes related to environmental factors [8].

Existing studies about dam anomaly detection generally follow a unified approach, entailing the creation of a predictive model to establish a baseline series. This is followed by calculating the residuals between the baseline and observed series and setting confidence intervals to identify anomalies. For instance, Li et al. [8] introduced an anomaly identification and warning system for dams using M-robust regression methods. This was further refined by Han et al. [9], who enhanced the M-robust linear regression technique and developed an efficient method for the online identification of anomalies in monitoring data. Xu et al. [10] devised a model to pinpoint anomalies in dam monitoring data, introducing a three-stage online process for outlier differentiation. Zhang et al. [11] proposed a data type-based self-matching model aimed at detecting anomalies in dams, addressing the limitations of single-method approaches to outlier identification. Despite the significant role these traditional methods play in processing dam monitoring data, they often struggle to capture the nonlinear relationships between series, particularly when dealing with data such as deformation sequences [12].

With advancements in machine learning (ML), the deployment of ML algorithms for DHM, particularly in anomaly detection, has received heightened interest. The application of ML in this context is divided into the following three categories based on data labeling: supervised learning, which employs classifiers trained on extensive labeled datasets to detect anomalies; unsupervised learning, which uses models trained on unlabeled data encompassing both normal and anomalous conditions to identify outliers; and semi-supervised learning, which begins with model training on unlabeled data, followed by refinement with a limited amount of labeled data for enhanced anomaly detection accuracy.

Recently, anomaly detection methods based on supervised learning have emerged. Salazar et al. [13] implemented anomaly detection in dam monitoring data using reinforced regression tree models and compared the performance of causal models, non-causal models, and autoregressive models. They emphasized the interpretability benefits of causal and non-causal models, alongside the simplicity and efficiency offered by autoregressive models. Further research by Salazar et al. [14] investigated anomaly detection within DHM using random vector machines and random forests, discussing the potentials and limitations of multi-class, two-class, and one-class classifiers. Despite the high accuracy of these methods, their development faces significant challenges, including managing large deformation monitoring datasets, difficulty in data labeling, and the scarcity of adequate training samples for supervised learning.

Conversely, unsupervised learning techniques, which do not necessitate labeled data, encompass a wide array of anomaly detection algorithms, such as clustering-based, distance-based, density-based, and prediction model-based strategies [15,16,17,18,19,20,21]. Researchers have explored the application of unsupervised learning techniques in detecting anomalies in DHM. Shao et al. [22] introduced a general and robust method for anomaly detection from the perspectives of image processing and artificial intelligence. Rong et al. [23] innovated a multipoint anomaly identification model, integrating an enhanced local outlier factor with mutual verification to account for spatiotemporal correlations. Ji et al. [24] introduced an anomaly detection strategy based on refined spectral clustering. Su et al. [25] introduced a diagnostic approach for dam structural behavior that combines probabilistic reasoning and data fusion. Dong et al. [26] presented a monitoring data anomaly identification method using an improved cloud model and radial basis function neural network. Liu et al. [27] developed an arch dam deformation anomaly detection model based on long short-term memory networks. These unsupervised approaches mitigate the challenge of label scarcity encountered in supervised methods for dam anomaly detection. Nonetheless, their effectiveness is significantly dependent on the precision of the models used [28], which limits their application. In response to these limitations, researchers have advanced the use of variational autoencoders (VAEs) [29] in anomaly detection, offering a promising direction for overcoming the obstacles associated with accurate modeling.

VAEs have become a prominent unsupervised learning technique in dam anomaly detection, focusing on training classifiers to learn the probability distribution of normal operational states. Zhou et al. [30] developed an innovative model that merges generative adversarial networks (GANs) with VAEs. Shu et al. [31] proposed a cutting-edge anomaly assessment framework based on sequential variational autoencoders coupled with Evidence Theory, enabling both the detection and fusion of anomalies. Subsequently, Shu et al. [32] further integrated spatiotemporal correlations into their model, employing temporal VAEs and graph convolutional neural networks for enhanced anomaly detection. While VAE-based models exhibit commendable performance in identifying anomalies in dam monitoring data, they often presuppose a Gaussian distribution of the underlying data, which is a presumption that may not accurately reflect the true distribution of dam deformation data. This theoretical discrepancy can affect the accuracy of reconstructing monitoring data. Furthermore, these methods are generally trained on normal data sequences, which is a practice that overlooks the presence of random anomalies in sensor-collected data, thereby complicating data preprocessing and diminishing the methods’ overall effectiveness and applicability. Despite these challenges, semi-supervised learning, which bridges the gap between supervised and unsupervised learning by potentially enhancing the accuracy of anomaly detection without requiring extensive labeled data, presents a viable alternative. However, the adoption of semi-supervised learning techniques within the DHM sector is still nascent, with more common use in sectors like mechanics [33] and environmental studies [34].

Addressing the limitations of current data-driven approaches for anomaly detection in dam deformation, this study identifies the following three primary challenges: (1) the dependency of supervised learning on labeled data and complex preprocessing, which restricts its applicability in large-scale engineering projects; (2) the constraints of unsupervised learning methods due to theoretical inaccuracies and reliance on the quality of datasets, affecting their precision; and (3) the prevalent focus on temporal characteristics of deformation, often neglecting or merely qualitatively analyzing spatial correlations, thereby missing out on the benefits of a detailed spatial analysis for accuracy improvement. To address these issues, this paper proposes an efficient anomaly detection method for dam deformation based on self-supervised learning. This novel approach comprehensively considers both the spatial correlations and temporal attributes of dam deformation, aiming to overcome the limitations associated with label generation in supervised learning and the accuracy dependence and theoretical discrepancies characteristic of unsupervised learning. For details on self-supervised learning, see Section 2.1.

The proposed methodology constructs a correlation matrix incorporating spatial associations and employs sliding window techniques to generate time sequences. It integrates convolutional blocks and transformer technology for effective information extraction and applies spatiotemporal contrastive learning to pre-train the encoder. This enables the distinguishing of unique dataset representations, followed by classifier fine-tuning for anomaly discrimination. The efficacy of this methodology is demonstrated through a case study. The primary contributions of this research are summarized as follows: (1) It proposes a self-supervised spatiotemporal contrastive pretraining (STCLP) method for representation learning in dam health monitoring, which, to the best of the authors’ knowledge, represents the first application of contrastive learning in this domain. (2) It proposes an anomaly detection method for dam deformation based on STCLP, which leverages large unlabeled data to enhance generalization performance, offering more timely and robust detection than other state-of-the-art techniques. (3) It improves the integration of spatial relationships by incorporating spatial contrastive learning on top of temporal ones, fully utilizing the spatial features of dam deformation for information extraction, thereby increasing the accuracy and rationale of the method.

This paper is structured as follows: Section 2 presents the background knowledge and implementation details of the proposed dam anomaly detection method. Section 3 details the case study analysis and discussion of results. Finally, Section 4 concludes this study and suggests directions for future research.

## 2. Methodology

### 2.1. Self-Supervised Learning and Contrastive Learning

Self-supervised learning [35], a variant of semi-supervised learning techniques, leverages custom-generated pseudo-labels for supervisory signals, thereby obviating the need for extensive manually annotated datasets. This approach facilitates the application of learned representations to a variety of downstream tasks. It can be divided into the following two categories [36]: generative [20,37] and discriminative [38,39]. Generative approaches aim to comprehend the underlying data distribution to produce outcomes that mimic real data closely. Conversely, discriminative techniques strive to differentiate among data variations, thus enabling precise input classification. While generative strategies are adept at capturing the data’s global attributes, discriminative methods excel in identifying local features and discrepancies within the input, making them particularly effective for sequence learning tasks that encompass a broad range of input variations, including both normal and anomalous data.

Contrastive learning is a form of discriminative self-supervised learning that brings the representations of similar samples (positive samples) closer together while pushing the representations of dissimilar samples (negative samples) apart, thereby enabling the model to capture essential features of the data. This goal is achieved by measuring the closeness of two embeddings through similarity metrics. Noteworthy contrastive learning models, such as SimSiam [40], SwAV [41], and SimCLR [42], have found substantial applications in image recognition. Building on the research of TS-TCC [43], a framework for learning representations of time series data through temporal and contextual contrast, this study proposes an advanced spatiotemporal contrastive learning framework. This framework is specifically designed to enhance anomaly detection in dam deformation by effectively capturing and differentiating between the nuanced spatial and temporal variations inherent in the data.

### 2.2. Implementation Details

This section introduces a method for dam anomaly detection using spatiotemporal contrastive learning. As illustrated in Figure 1, the comprehensive framework encompasses the following three pivotal stages: data acquisition and preprocessing, self-supervised pretraining, and fine-tuning coupled with anomaly detection. Initially, during the data collection and preprocessing stage, dam deformation data are automatically gathered from the dam’s pendulum monitoring system. The data are preprocessed to construct the dataset, which is subsequently segmented into training, validation, and testing subsets in predefined ratios, readying it for model ingestion. In the self-supervised pretraining stage, the method leverages transformer architectures alongside nonlinear projection heads to extract spatiotemporal features from the datasets. These features undergo pretraining through a spatiotemporal contrastive training strategy, optimizing the model to recognize pertinent data characteristics. The final stage, fine-tuning and anomaly detection, involves adapting the pretrained model parameters for specific downstream applications. Here, the model is refined using a minimal set of labeled data, after which it is deployed for the task of anomaly detection, thus concluding the process.

#### 2.2.1. Data Processing

The aforementioned method requires the construction of a dataset containing spatial and temporal features to facilitate model input. Given the premise that a dam experiences comparable forces and environmental influences, it is hypothesized that the deformation at any given monitoring point is interrelated with its neighboring points. When a local anomaly occurs in a certain part of the dam, the likelihood of anomalies in its surrounding position increases. Consequently, the spatial features of dam deformation can be collectively determined by the deformation variables of a measuring point and its surrounding points. Utilizing the dam’s pendulum monitoring system, the deformation field of the dam can be represented by the time sequences of multiple measuring points, capturing the correlations and dynamic changes among deformations at different dam locations, it can be expressed as follows:(1)$$Y=\left(\begin{array}{ccc} y_{11} & \ldots & y_{1n}\\ \vdots & \ddots & \vdots \\ y_{m1} & \cdots & y_{mn} \end{array}\right)\left(i=1,2,\ldots ,m;t=1,2,\ldots ,n\right)$$

Here, $y_{it}$ represents the measured deformation [44], *m* is the total number of monitoring points, and *n* is the monitoring period.

However, deformation patterns vary across different dam regions, necessitating a nuanced approach to constructing spatial features that considers the varying correlation of deformation characteristics. Previous methods employed Pearson correlation analysis [45] to assess the relationship between the prediction target and variables at different locations, which is as follows:(2)$$corr_{ij}=\frac{\sum_{t=1}^{n}\left(y_{it}-{\bar{y}}_{i}\right)\left(y_{jt}-{\bar{y}}_{j}\right)}{\sqrt{\sum_{t=1}^{n}{\left(y_{it}-{\bar{y}}_{i}\right)}^{2}{\left(y_{jt}-{\bar{y}}_{j}\right)}^{2}}}\left(i,j=1,2,\ldots ,m;t=1,2,\ldots ,n\right)$$

These methods applied a threshold *S* to filter correlations, excluding all inputs in which the correlation coefficient fell below *S*. While this approach differentiated between correlated and uncorrelated measurement points, it did not account for the degree of association between variables at different locations. This paper proposes an improved method that builds upon the previous approach by introducing a correlation matrix *A*, as follows:(3)$$A=\left(\begin{array}{ccc} corr_{11} & \ldots & corr_{1m}\\ \vdots & \ddots & \vdots \\ corr_{m1} & \cdots & corr_{mm} \end{array}\right)\left(i,j=1,2,\ldots ,m\right)$$

The essence of this matrix lies in its ability to transform the original measurement data *Y* into a new matrix $\hat{Y}$ through multiplication with *A*. This resultant matrix $\hat{Y}$ assimilates insights by accounting for the degrees of variable association across various locations. This advanced approach ensures a thorough consideration of the interrelationships between locational variables, transcending mere threshold-based filtering. Such improvements afford a more precise capture and utilization of spatial features, providing stronger support for subsequent model training and prediction. After this, sliding window techniques are employed to slide the information in the matrix along the time axis, constructing continuous time series sequences with spatial information to form the dataset.

#### 2.2.2. TSCLP for DHM

In this section, a self-supervised pretraining method, TSCLP, that is suitable for DHM is proposed through comparative learning. Figure 2 depicts the diagram of the proposed spatiotemporal contrastive learning training method. This method enhances temporal contrastive learning [43] by integrating spatial correlations pertinent to dam deformation, thereby making it more effective for dam health monitoring. The method unfolds in the following three stages: data augmentation, temporal contrastive learning, and spatial contrastive learning modules. By separately employing temporal and spatial contrastive techniques, the method captures the independent yet complementary characteristics of both dimensions. Temporal contrast improves the ability to identify critical changes over time, while spatial contrast highlights regional patterns and deformations across the structure. Each stage is described in detail in the following sections.

##### Stage 1: Data Augmentation

Data augmentation plays a crucial role in the efficacy of contrastive learning, far surpassing its importance in supervised learning [46]. There are various signal processing methods available for 1D signal processing, such as Gaussian noise, amplitude scaling, and time stretching. Typically, contrastive learning methods utilize identical data augmentation techniques to construct similar feature samples, but employing varied augmentations can enhance the robustness of feature learning.

In this context, both weak and strong data augmentation methods are applied. For a given time series $X=\left[x_{1},x_{2},\cdots ,x_{n}\right]$, weak augmentation involves scaling the input time series data by multiplying each channel by a random factor drawn from a normal distribution, introducing scale variations.
(4)$$X\prime =X\cdot \left(1+\epsilon \right)$$
where ϵ represents the scaling factor.

Conversely, strong augmentation incorporates permutations and jitter, slightly rearranging the time steps in the first channel of each sample and injecting random noise from a normal distribution into each time step.
(5)$$X\prime \prime =\mathrm{Permute}\left(X\right)+\eta$$
where Permute (X) denotes the random permutation of the time steps of X, and *η* represents random noise that follows a normal distribution.

These data augmentation operations contribute to improving the model’s generalization ability, enabling it to handle unseen data variations more effectively during training. As illustrated in Figure 3, The original data are a periodic time series with missing values. After weak augmentation, the scale of the series is doubled, while the rest remains unchanged; after strong augmentation, the time steps of the series are randomly permuted, and random noise is added, resulting in a transformed series without missing values.

##### Stage 2: Temporal Contrast Module

The temporal contrast module is integral to contrastive learning, which emphasizes discerning similarities and differences among samples to derive representations. This process is facilitated through a contrastive loss function, which is predicated on the idea that samples belonging to the same category should closely resemble each other, while those from different categories should not. By integrating contrastive loss with an autoregressive model, this module effectively extracts temporal features within a latent space. Following the application of weak and strong augmentations, samples are processed by an encoder comprising three convolutional blocks designed to distill local information from the augmented data into high-dimensional latent features. A transformer, recognized for its efficiency and rapid processing, serves as the autoregressive model to capture temporal information, with a sequential architecture [47] primarily consisting of multi-head attention and MLP blocks. These MLP layers feature two fully connected layers, nonlinear activation functions, and dropout mechanisms, enhanced by residual connections to stabilize gradients and optimize temporal analysis. This process involves a cross-view prediction task, in which information from strongly augmented samples at the current time step is used to predict the features of weakly augmented samples at future time steps and vice versa. The goal of the temporal contrastive loss is to minimize the dot product between the predicted representation and the true representation of the same sample while maximizing the dot product with other samples within the same batch.

##### Stage 3: Spatial Contrast Module

Temporal information alone is insufficient for analyzing spatially correlated time series data, such as those found in dam deformation. The spatial contrast module, when added to the temporal–spatial contrastive learning pretraining (TSCLP) framework, enables the extraction of spatiotemporal features. Utilizing contrastive loss alongside nonlinear transformations, this module extracts spatial features by transforming predictions from the temporal contrast module through nonlinear projection heads, which facilitate spatial comparisons. These projection heads consist of two linear layers, including a batch normalization layer and a nonlinear activation function, allowing for the encapsulation of spatial details across multiple original data samples. Each set of samples, modified through weak and strong augmentations and transformations, yields two sets of spatial features. Spatial features generated from two augmented views of the same input are considered a pair of positive samples, while those from different inputs within the same batch are treated as a pair of negative samples, as illustrated in Figure 4. The spatial contrastive loss function aims to maximize the similarity between pairs of positive samples and minimize the similarity between pairs of negative samples, thereby ensuring the final representation captures spatial correlations.

#### 2.2.3. Implementation Procedures of TSCLP

The TSCLP training process is outlined in Algorithm 1. For each input sample x, weakly augmented sequences $x^{w}$ and strongly augmented sequences $x^{s}$ are obtained through data augmentation. An encoder then extracts high-dimensional information to produce d-dimensional latent features z, as follows:(6)$$z=\mathrm{fenc}\left(x\right),z\in R^{d}$$

This process results in feature representations $z^{w}$ and $z^{s}$ for weakly and strongly augmented sequences, respectively. The subsequent step involves the temporal contrast phase, in which an autoregressive analysis, executed via a transformer model, generates $c_{t}$ as follows:(7)$$c_{t}=f_{\mathrm{tran}}\left(z\leq t\right),c_{t}\in R^{h}$$
where h represents the hidden dimension of the transformer module.

Then, xt+k and ct are operated by the log-bilinear model, yielding the future time step features zt+k, as follows:(8)fkxt+k,ct=expWkckTzt+k,Wk:ℝRh→d
where Wk represents a linear function that aligns ct to the dimensionality of z. This process generates two sets of temporal features, ctw and cts, using cross-view prediction.

The core objective of the training is to minimize the dot product between the true and predicted values for identical samples while amplifying the dot product for different samples within the same batch. This aim is encapsulated within the temporal loss function as follows:(9)$$L_{TC}^{s}=-\frac{1}{K}\sum_{k=1}^{K}\log\frac{\exp\left({\left(W_{k}\left(c_{t}^{s}\right)\right)}^{T}z_{t+k}^{w}\right)}{\sum n\in N_{t,k}\exp\left({\left(W_{k}\left(c_{t}^{s}\right)\right)}^{T}z_{n}^{w}\right)}$$
(10)$$L_{TC}^{w}=-\frac{1}{K}\sum_{k=1}^{K}\log\frac{\exp\left({\left(W_{k}\left(c_{t}^{w}\right)\right)}^{T}z_{t+k}^{s}\right)}{\sum n\in N_{t,k}\exp\left({\left(W_{k}\left(c_{t}^{w}\right)\right)}^{T}z_{n}^{s}\right)}$$

After this, the spatial contrast phase involves the transformation of ctw and cts via nonlinear projection, resulting in two spatial feature sets, Ow and Os, with the number of features in O equaling the number of input samples N, resulting in 2N spatial features. For any given feature Oi, the pair Oiw,Ois is treated as a positive sample, whereas the other 2N−2 spatial features from other inputs in the same batch are deemed negative samples for Oiw. The goal of this phase is to maximize the similarity of positive sample pairs and minimize the similarity of negative sample pairs, simplifying the division of the similarities with the spatial loss function, as follows:(11)LSC=−∑i=1Nlogexpsimoi,tw,oi,ts/τ∑m=12Nℚm≠iexpsimoi,tw,oi,ts/τ
where Qm≠i is an indicator function that is assigned a value of 1 when m≠I; τ represents a temperature parameter; and *sim* denotes the similarity calculation simυ,ν=υTν/υν.

Then, the model’s overall loss is a weighted sum of temporal and spatial losses, which can be described as follows:(12)$$L=\lambda_{1}\cdot \left(L_{TC}^{s}+L_{TC}^{w}\right)+\lambda_{2}\cdot L_{SC}$$
where $\lambda_{1}$ and $\lambda_{2}$ are weights. Through this training, the model acquires sequences enriched with spatiotemporal information.
**Algorithm 1:** temporal and spatial contrast training.Input: sample x, constant t, k, n, weight λ1, λ2, 1 Randomly initialize model parameters$2\;\mathrm{for}\;\mathrm{all}\;i=\left\{1,2,\ldots ,n\right\}$ do:3   //data augmentation$4\;\;\;x_{i}^{w}=\mathrm{weak}\_\mathrm{augmentation}(x_{i})$$5\;\;\;x_{i}^{s}=\mathrm{strong}\_\mathrm{augmentation}(x_{i})$6   //Forward calculation$7\;\;\;z_{i}^{w}=\mathrm{Encoder}\left(x_{i}^{w}\right)$$8\;\;\;z_{i}^{s}=\mathrm{Encoder}\left(x_{i}^{s}\right)$$9\;\;\;\mathrm{for}\;\mathrm{all}\;t=\left\{1,2,\ldots ,k\right\}$ do:$10\;\;\;\;\;\;c_{i,t}^{w}=\mathrm{Transformer}\left(z_{i,t}^{w}\right)$$11\;\;\;\;\;\;c_{i,t}^{s}=\mathrm{Transformer}\left(z_{i,t}^{s}\right)$$12\;\;\;\;\;\;\mathrm{Get}\;\mathrm{temporal}\;\mathrm{loss}\;L_{TC}^{w}$$\;\mathrm{by}\;c_{i,t}^{w}$$,\;z_{i,t+k}^{s}$$\;\mathrm{and}\;L_{TC}^{s}$$\;\mathrm{by}\;c_{i,t}^{s}$$,\;z_{i,t+k}^{w}$13   end$14\;\;\;o_{i}^{w}=NPH\left(c_{i}^{w}\right)$$15\;\;\;o_{i}^{s}=NPH\left(o_{i}^{w}\right)$16 end$17\;\mathrm{Get}\;\mathrm{spatial}\;\mathrm{loss}\;L_{SC}$$\;\mathrm{by}\;\mathrm{positive}\;\mathrm{sample}\;o_{i}^{w}$$,\;o_{i}^{s}$ and negative sample in x$18\;\mathrm{Get}\;\mathrm{total}\;\mathrm{loss}:\;L=\lambda_{1}\cdot \left(L_{TC}^{w}+L_{TC}^{s}\right)+\lambda_{2}\cdot L_{SC}$19 Update model by L  Output: Encoder parameters

#### 2.2.4. Fine-Tuning and Anomaly Detection

In unsupervised anomaly detection tasks, the presence of outliers can interfere with model training and adversely affect detection accuracy. To mitigate this, a semi-supervised fine-tuning approach is employed. As illustrated in Figure 1, after dataset construction, the TSCLP model undergoes unsupervised pretraining on the unlabeled dataset, with the encoder’s parameters being saved upon training completion. Subsequently, these parameters are transferred to downstream tasks, in which the encoder is fine-tuned in a supervised manner using a small, labeled dataset curated based on expert experience. After fine-tuning, the network’s updated parameters are saved. Finally, the fine-tuned network is applied to the test dataset for anomaly detection, yielding the final detection outcomes.

### 2.3. Performance Evaluation Metrics

This study evaluates the model’s performance using the following four widely recognized metrics: accuracy, precision, recall, and *F*_1_ score. These metrics are essential for a comprehensive assessment, each offering insight into different aspects of the model’s effectiveness across positive and negative classifications. Accuracy represents the proportion of correctly predicted observations (encompassing both true positives and true negatives) to the total observations, offering a broad view of the model’s overall performance. Precision is the ratio of correctly predicted positive observations to the total predicted positives, reflecting the model’s precision in identifying positive classes. Recall, or sensitivity, determines the proportion of correctly predicted positive observations to all actual positives, evaluating the model’s ability to detect all pertinent instances. The *F*_1_ score, the harmonic mean of precision and recall, provides a balanced measure that accounts for both precision and recall. The formulas for these metrics are as follows:(13)$$Accuracy=\frac{TP+TN}{all}$$
(14)$$Precision=\frac{TP}{TP+FP}$$
(15)$$Recall=\frac{TP}{TP+FN}$$
(16)$$F_{1}=\frac{2\times precision\times recall}{precision+recall}$$
where true positives (*TP*) denotes the count of anomalies accurately identified, false positives (*FP*) signifies the count of non-anomalies erroneously classified as anomalies, false negatives (*FN*) represents the count of anomalies that were overlooked, true negatives (*TN*) refers to the count of non-anomalies correctly identified as such, and all is the total number of data points.

## 3. Case Study

The proposed anomaly detection method was evaluated using real-world engineering data on horizontal deformation. The development and testing of the models took place in a Python 3.7 and PyTorch 1.7 environment on a computer configured with an Intel(R) Core(TM) i7-8700K CPU at 3.70 GHz and an NVIDIA GeForce RTX 2080 Ti graphics card with 11 GB of VRAM.

### 3.1. Data Collection and Processing

The case study focuses on a concrete arch dam in Yunnan Province, China. Figure 5 presents photographs of the dam from upstream and downstream perspectives. Horizontal deformation, a critical monitoring aspect, is tracked by a pendulum system installed within the dam structure. Figure 6 illustrates the comprehensive arrangement of the dam’s pendulum system. In seven dam sections (9#, 15#, 19#, 22#, 25#, 29#, and 35#) across various gallery levels (at elevations of 1014 m, 1054 m, 1100 m, 1150 m, and 1190 m), vertical pendulums are strategically placed in segments within the foundation gallery, grouting gallery, and at the inverted pendulum connections to monitor the dam’s horizontal deformation and deflection. The pendulums primarily employ automated monitoring with the use of capacitive pendulum inclinometers, recording observations three times a day. For additional accuracy, manual observations using optical pendulum inclinometers are conducted monthly.

Figure 7 presents the long-term monitoring series of horizontal deformations at various measuring points, with the sample series spanning from 1 January 2012, to 3 December 2018, at a daily data sampling frequency. An analysis reveals that changes in upstream water levels predominantly influence the dam’s horizontal deformation, resulting in downstream deformation as water levels increase and an upstream rebound when they decrease. Moreover, deformation trends and periodic changes at different locations exhibit remarkable similarity, and nearly all measuring points show anomalies around 1 January 2013, indicating spatial correlation among the deformations. The collected data contain missing values and outliers and involve numerous measuring points, making label creation time-consuming and labor-intensive, thereby rendering supervised learning impractical.

A Pearson correlation analysis was performed on the deformation data from each measuring point, with the results displayed in Figure 8. The color gradient from white to blue indicates the absolute value of the correlation coefficient, ra nging from 0 to 1, where asterisks denote significance levels. The analysis reveals that all measuring points are correlated, with correlation coefficients above 0.5 and significance values (*p*) less than 0.001, suggesting that the correlations between measuring points are statistically significant. In this study, a threshold (S) of 0.6 was selected; points with correlations above this threshold were considered to have significant relationships. The original data were then multiplied by the corresponding correlation matrix to generate a sequence with spatial features. This sequence underwent initialization and sliding window operations to create a dataset with spatiotemporal features, which was divided into training, testing, and validation sets in a 4:1:1 ratio.

### 3.2. Implementation Details

After dividing the dataset, fivefold cross-validation was performed using a network search with different random seeds to identify the optimal hyperparameters for the network’s pretraining and downstream tasks, as detailed in Table 1. The Adam optimizer was employed for both pretraining and downstream tasks due to its capacity for faster convergence when training neural networks. During pretraining, the model underwent 100 epochs, with an early stopping algorithm implemented to halt training when optimal performance was achieved. This optimization revealed that model performance generally peaked around 40 epochs, which was then adopted for the downstream tasks. After extensive experimentation and comparison, the optimal weights for temporal and spatial loss were found to be $\lambda_{1}=1$ and $\lambda_{2}=0.7$, respectively; the process is elaborated in the subsequent sensitivity analysis of parameters.

Furthermore, in the parameter settings for data augmentation, a scaling factor of 1.1 was used during the weak augmentation phase, while the maximum number of fractional parts in strong augmentation was set to five, and the standard deviation parameter for random noise was set to 0.1 to enhance the diversity of the dataset. Such parameters are designed to foster the model’s capacity to learn robust features amidst diverse and noisy data. As for the transformer network’s configurations, the hidden dimension was established at 100, with the network comprising four attention layers, each with four heads, and the multilayer perceptron’s hidden layer dimension set at 64. These configurations aim to provide sufficient model complexity to effectively capture the temporal and spatial dependencies of the input sequences.

### 3.3. Anomaly Detection

In this setup, the model is initially pretrained on a selection from the unlabeled training dataset. It is then fine-tuned on a small dataset to which labels have been randomly assigned. Due to the limited size of the fine-tuning dataset, data augmentation techniques are employed to enhance data diversity. Taking A09-PL-01 as an example, the processes of pretraining and fine-tuning are illustrated in Figure 9. The figure demonstrates that during pretraining, the loss decreases with an increase in the number of training epochs, stabilizing around 40 epochs. In the fine-tuning phase, the loss continues to decrease, with accuracy progressively exceeding 0.9. After fine-tuning, the model’s ability to detect anomalies in the unlabeled test set is evaluated. The model’s predictions are output as probability values, with probabilities above a certain threshold indicating positive classes (anomalies) and those below it indicating negative classes (non-anomalies).

### 3.4. Result Analysis

Following the previous description, the proposed method was applied to detect anomalies in the deformation data of 31 measuring points on the dam, with the results presented in Table 2. This table provides a detailed breakdown of the evaluation metrics during the training validation phase and the testing phase. Overall, the model demonstrated exceptional performance, achieving accuracy and precision rates above 95% and recall rates above 75%, with *F*_1_ scores exceeding 80% across most of the test dataset, except for a few measuring points. Reduced performance at certain locations, such as A09-PL-01 and A35-PL-02, is linked to an increased presence of outliers in the training set, which posed challenges to the unsupervised training process and negatively impacted the training efficiency and anomaly detection capabilities of the model.

Additionally, the table indicates a less consistent performance across measuring points during the testing phase compared to the training and validation phases. In some instances, the detection rate during the testing phase dropped below that of earlier phases, while in others, it reached 100%. A thorough analysis of the original data identifies the following underlying cause of this phenomenon: the random distribution of anomalous data throughout the dataset. Given the data length ratio of 4:1 between the training validation set and the test set, the latter has limited data, consequently reducing the number of anomalies. This results in more extreme detection outcomes under certain conditions.

Figure 10 presents the anomaly detection results for the test sets of seven representative measuring points. In the figure, black lines represent the data of the test set, red circles indicate correctly identified anomalies, blue squares denote anomalies that were not detected, and green triangles signify normal values incorrectly labeled as anomalies. Observation of the results shows that the proposed method accurately detects both missing and abrupt values, with instances of false positives and false negatives being exceedingly rare, underscoring the method’s exceptional performance.

### 3.5. Comparison with Other Methods

To evaluate the performance of the proposed anomaly detection method, it was benchmarked against the following three leading unsupervised learning approaches: temporal contrastive learning (TC), transformer-based anomaly detection (TAD), and generative adversarial network variational autoencoder (GAN-VAE). The TC method relies on a temporal contrast model for training and a small dataset for fine-tuning, employing a classification task for anomaly detection. The proposed method expands upon TC by integrating spatial correlations and utilizing spatial contrast for improved performance. The TAD leverages a transformer to reconstruct normal sequences and calculates anomaly scores based on reconstruction errors for anomaly detection. Similarly, transformers are used in the proposed method to extract sequence information. GAN-VAE models combine the adversarial learning power of GANs and the probabilistic generative modeling of VAEs, constructing samples that closely resemble the original data’s distribution and detecting anomalies through the relationship between reconstruction errors and thresholds. Unlike the generative learning approach of GAN-VAE models within the unsupervised learning paradigm, the proposed method falls under the contrastive learning branch of unsupervised learning. The proposed method focuses specifically on displacement analysis; therefore, Zhou et al.’s information fusion component was not included in the calculations.

Similarly, the dataset was divided into training, testing, and validation sets with a ratio of 4:1:1. The parameter configurations for the three methods are detailed in Table 3. The parameters for the TC method are identical to those of the proposed method. In the TAD method, the number of attention heads is four, and the size of the hidden layer is 32. For the GAN-VAE method, both the encoding and decoding phases use a one-dimensional convolutional neural network with 32 convolutional kernels and a kernel size of seven. The learning rates are set at 0.001 for the VAE and generator and 0.0002 for the discriminator. A KL loss weighting coefficient of 0.1 is applied, with an additional GAN loss coefficient of 0.5. The noise dimension is set to 100 for GAN training. The threshold values for the TAD and GAN-VAE methods are set to 10% of the data range.

Figure 11 displays the comparative results of anomaly detection across various methods for representative measuring points. The four axes represent precision, recall, accuracy, and *F*_1_ score metrics, with identical value ranges and scale sizes across axes. A metric value closer to one indicates better detection performance. The shaded areas in different colors illustrate the performance of each method on these metrics, as follows: red for the proposed method, blue for TC, green for TAD, and black for GAN-VAE. The size of the shaded area visually represents the method’s overall effectiveness, with larger areas indicating superior anomaly detection capabilities.

Overall, the red-shaded area completely encompasses the other colored areas, indicating that the proposed model outperforms the other models across all measuring points. The shape of the red area is nearly square, signifying that the proposed method demonstrates balanced performance across all four metrics, ensuring stable anomaly detection capabilities. Although the accuracy values of each method reach around 90%, with minor differences, significant disparities exist in the other three metrics. This is because the accuracy metric reflects the overall detection accuracy across all data, including both normal and anomalous values. The high proportion of normal values in the original data suggests that all models exhibit strong detection capabilities for normal values.

Specifically, both the TC and TAD methods exhibit advantages and outperform the GAN-VAE method. Except for A09-PL-01, the TC method consistently achieves metric values above 0.7, indicating stable performance capabilities. In contrast, the TAD method demonstrates high precision values, yet its recall and *F*_1_ scores are comparatively lower, with instances in which precision exceeds recall by 0.6, signifying that while TAD accurately identifies anomalies, it also has a high rate of false positives. The GAN-VAE method shows inferior performance in precision, recall, and *F*_1_ scores, significantly influenced by its assumptions about data distribution, the selection of thresholds, and the inherent instability of adversarial training.

Although the TC method also employs contrastive learning for training, its metric values are approximately 0.2 lower than those of the proposed method across almost all indicators, which significantly highlights the importance of spatial correlations in the analysis of dam deformation. The proposed method enhances the TC approach by incorporating spatial contrast loss, thus offering a more comprehensive consideration of spatial correlations. Similar to the proposed method, the TAD approach processes temporal information using transformers, yet it exhibits a higher rate of missed detections, further demonstrating the superiority of contrastive training over conventional mean squared error (MSE) training.

In summary, the proposed method demonstrates exceptional performance in anomaly detection, particularly when accounting for spatial correlations, which further enhances its efficacy.

### 3.6. Sensitivity Analysis of Parameters

The efficacy of the proposed anomaly detection method is contingent upon the selection of time steps and the allocation of weights to temporal and spatial losses. This section provides a preliminary discussion on the choice of time steps and the values of loss weights $\lambda_{1}$ and $\lambda_{2}$. Taking the data from the A01-PL-09 measuring point as an example, anomaly detection was conducted within various ranges of time steps and weights. The preliminary patterns of these parameters were analyzed with precision as the evaluation metric.

Figure 12a illustrates the impact of time step selection on overall performance, in which the x-axis represents the proportion of time step length to feature length, and the y-axis shows the variation in precision values, with all weights set to 1.0. The graph indicates that moderately increasing the proportion of time steps can enhance performance, but an excessive proportion may impair it. This is because larger time steps reduce the dataset available for training, leading to poorer outcomes. In our dataset, the model performs best when the time step is approximately 30% of the feature length (batch size), thus setting the time step to 10 in the configuration.

Figure 12b displays the impact of contrast loss weights on overall performance. The horizontal axis shows the range of weight variations, and the vertical axis displays the accuracy values. Green markers denote the results of changing $\lambda_{1}$ while holding $\lambda_{2}$ at 1.0; orange markers indicate the effects of altering $\lambda_{2}$ with $\lambda_{1}$ fixed at 1.0. It was observed that the model exhibits optimal performance when $\lambda_{2}$ is 1 and $\lambda_{1}$ is 0.3. Moreover, the model appears to be more sensitive to variations in $\lambda_{1}$ than to changes in $\lambda_{2}$. This sensitivity can be attributed to the fact that, under real-world conditions, dam deformation is significantly influenced by seasonal variations, while the impact of spatial correlations within the dam structure is comparatively minor. Generally, the spatial relationships in dam deformation remain at a consistent level.

## 4. Conclusions

This study introduced a dam deformation anomaly detection method based on self-supervised spatiotemporal contrastive pretraining. It constructs temporal and spatial contrast modules that learn similar representations by maximizing the similarity of positive sample pairs and minimizing the similarity of negative pairs within each module. To extract temporal and spatial features between inputs, transformers and nonlinear projection heads were utilized. The framework includes dataset construction, unsupervised pretraining, parameter transfer, semi-supervised fine-tuning, and anomaly detection. The performance of this method was comprehensively studied, yielding the following conclusions:

The model exhibits excellent performance, achieving over 95% in both accuracy and precision, above 75% in recall, and over 80% in the *F*_1_ score, with only a few exceptional measuring points.

When applied to a real arch dam engineering case and compared with three benchmark models, the proposed method outperforms other unsupervised learning approaches. The analysis underscores the critical importance of spatial correlations in dam deformation analysis, with spatially aware methods showing superior anomaly detection outcomes than those that do not consider such correlations.

Sensitivity analysis of the model’s hyperparameters indicates that an appropriate increase in the time step ratio can enhance performance, whereas excessive time steps may impair it. The model is particularly sensitive to changes in $\lambda_{1}$ and less so to changes in $\lambda_{2}$.

Proving adept at identifying local changes in input data, the proposed method is not limited to deformation metrics but is also applicable to detecting anomalies in dam body stress, seepage, cracks, and other relevant data types. Future research will explore the application of this method in comprehensive anomaly detection for dams.

## Figures and Tables

**Figure 1 sensors-24-05858-f001:**
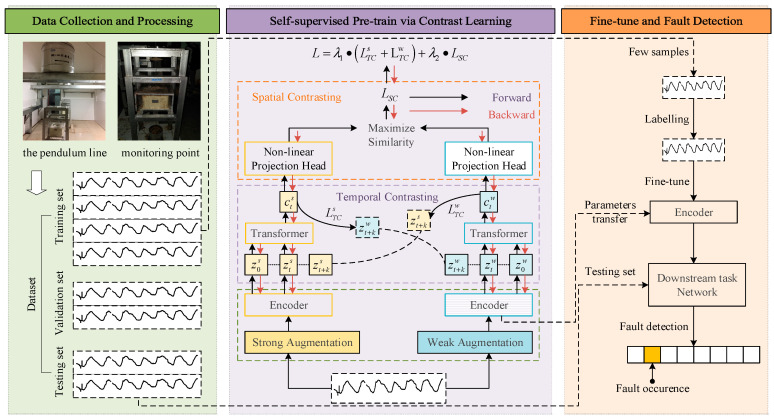
Overall framework of the proposed method.

**Figure 2 sensors-24-05858-f002:**
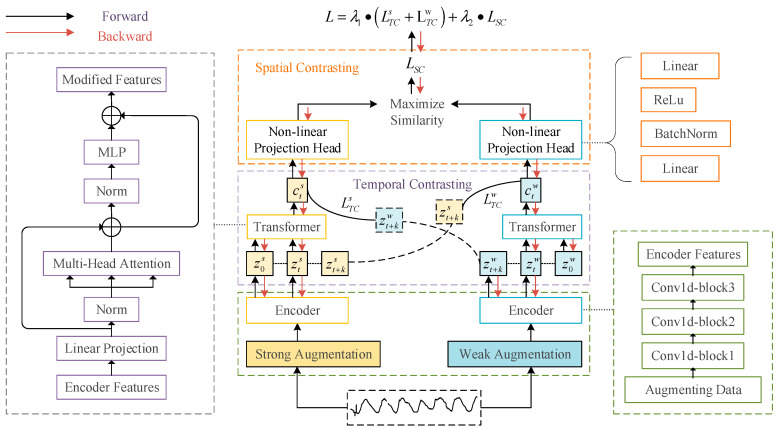
Diagram of TSCLP.

**Figure 3 sensors-24-05858-f003:**
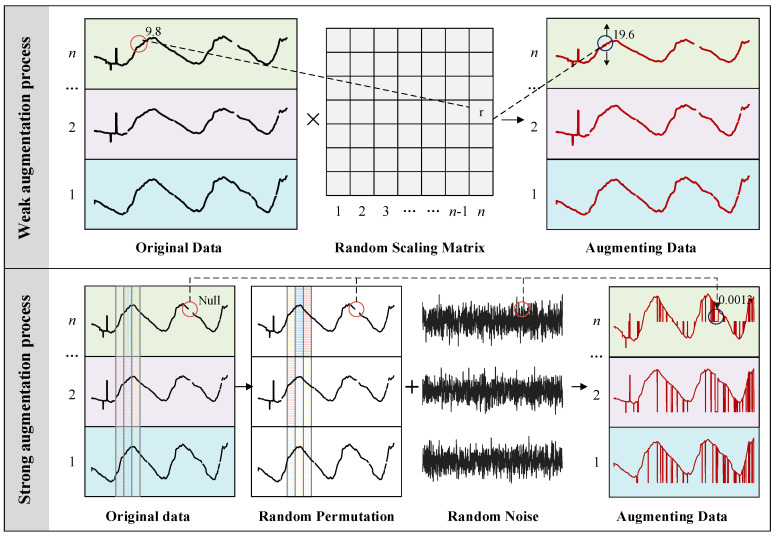
Diagram of data augmentation.

**Figure 4 sensors-24-05858-f004:**
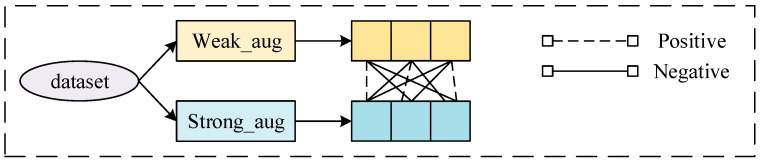
Definition of positive and negative samples.

**Figure 5 sensors-24-05858-f005:**
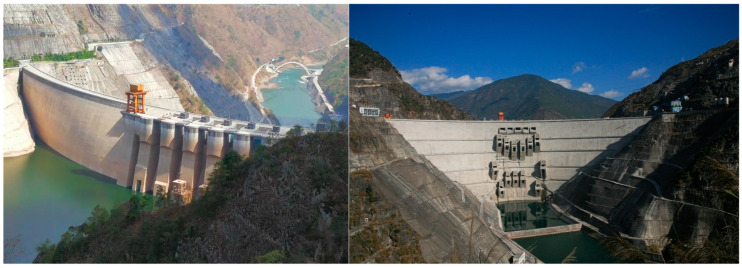
Perspectives of the arch dam.

**Figure 6 sensors-24-05858-f006:**
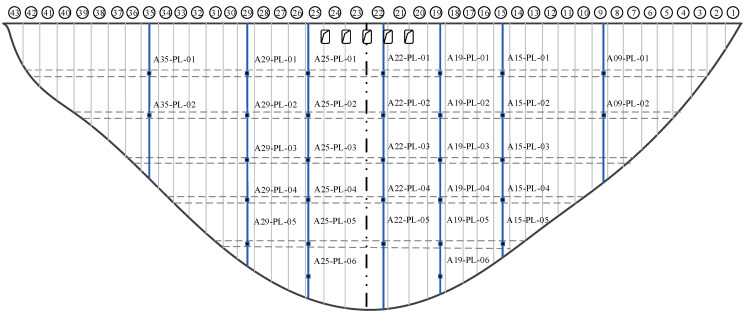
Pendulum systems for monitoring horizontal deformation.

**Figure 7 sensors-24-05858-f007:**
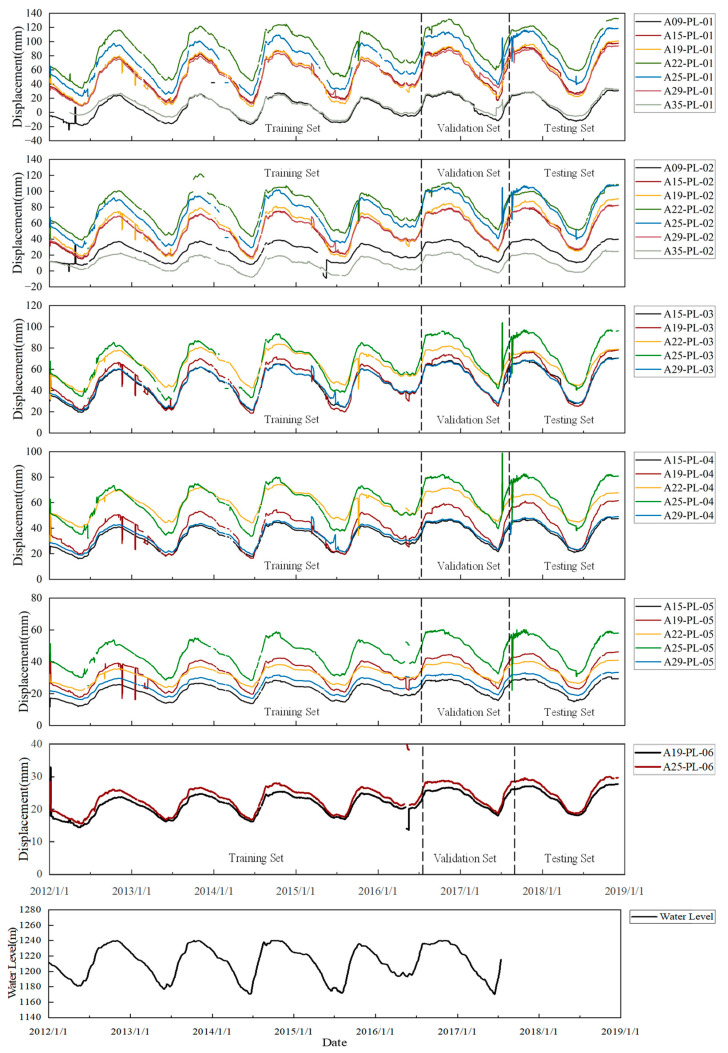
Deformation series of monitoring data for selected sensors.

**Figure 8 sensors-24-05858-f008:**
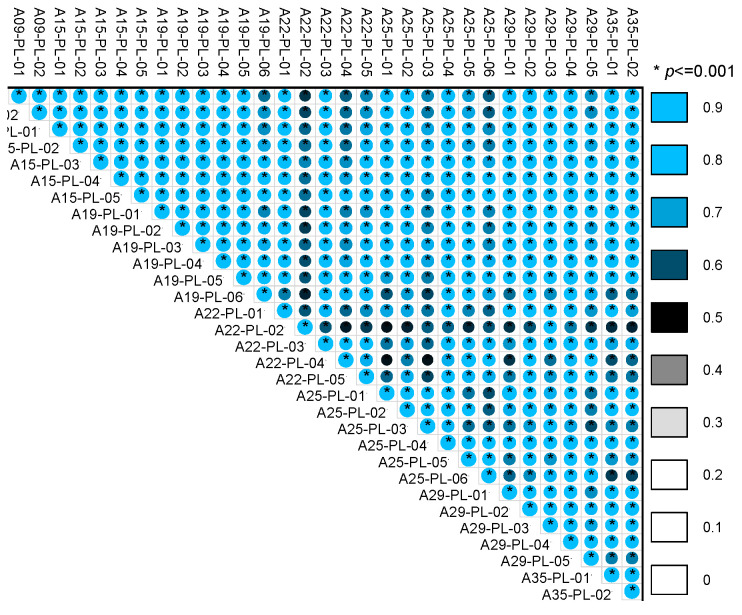
Matrix graph of the deformation correlation coefficient between different monitoring points.

**Figure 9 sensors-24-05858-f009:**
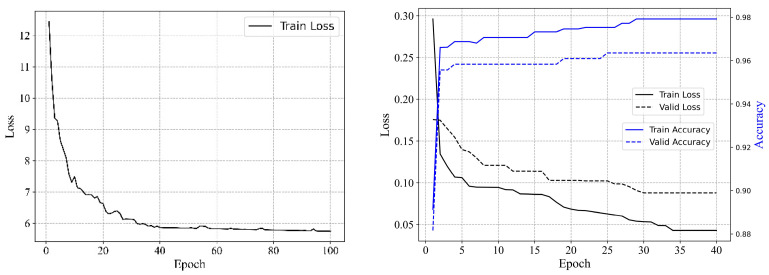
Loss and accuracy curves.

**Figure 10 sensors-24-05858-f010:**
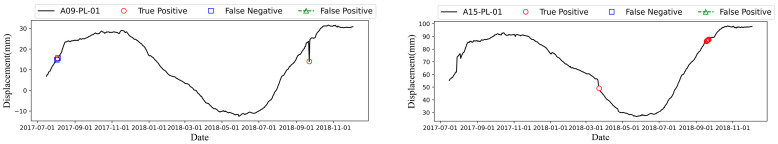
Anomaly detection results of seven representative measuring points.

**Figure 11 sensors-24-05858-f011:**
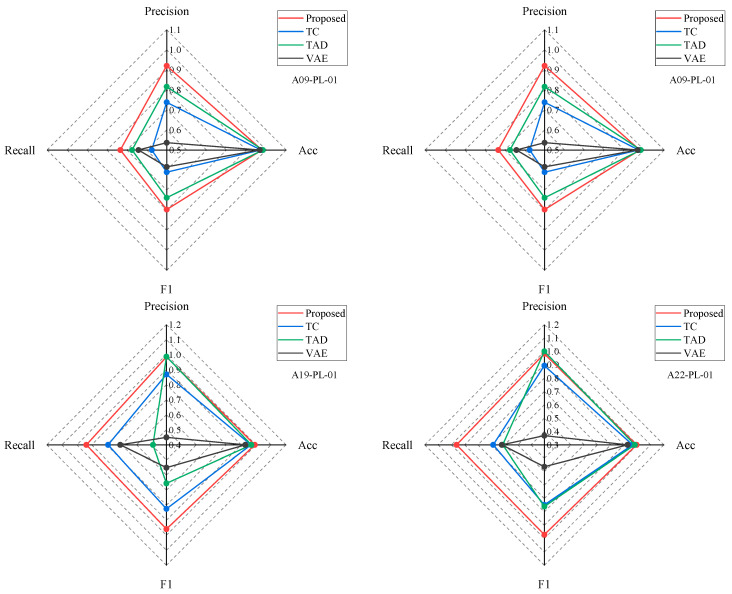
Comparison with other state-of-the-art methods.

**Figure 12 sensors-24-05858-f012:**
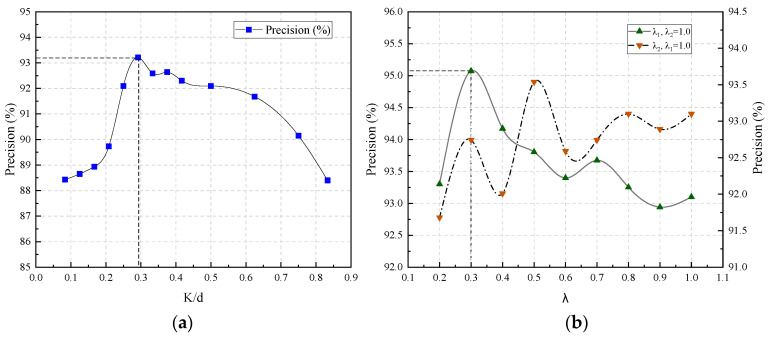
Sensitivity analysis results. (**a**) The impact of time steps on performance. (**b**) The impact of weight changes on performance.

**Table 1 sensors-24-05858-t001:** Hyperparameter setting after cross-validation.

Hyperparameter	Pre-Train	Downstream Task
Batch size	32	32
Optimizer	Adam	Adam
Learning rate	0.0001	0.0001
Weight decay	0.0001	0.0001
$\beta_{1}$	0.9	0.9
$\beta_{2}$	0.99	0.99
Epoch	100	40
Timestep	10	10
$\lambda_{1}$	1	-
$\lambda_{2}$	0.7	-
τ	0.2	-

**Table 2 sensors-24-05858-t002:** Performance of the proposed method.

Measurement Points	Train Set and Valid Set (%)				Test Set (%)			
	Acc	Precision	Recall	$F_{1}$	Acc	Precision	Recall	$F_{1}$
A09-PL-01	97.79	92.01	73.23	79.71	99.80	99.90	87.50	92.81
A09-PL-02	99.76	99.24	98.62	98.93	100.00	100.00	100.00	100.00
A15-PL-01	99.76	99.13	98.41	98.77	100.00	100.00	100.00	100.00
A15-PL-02	98.85	99.41	86.32	91.78	99.80	99.90	75.00	83.28
A15-PL-03	99.72	99.86	94.93	97.26	99.60	99.80	75.00	83.23
A15-PL-04	99.92	99.42	99.42	99.42	99.80	90.00	99.90	94.39
A15-PL-05	99.84	99.92	97.22	98.53	99.80	99.90	91.67	95.40
A19-PL-01	99.09	98.52	93.61	95.91	99.01	91.27	77.68	83.08
A19-PL-02	99.05	97.92	92.56	95.06	100.00	100.00	100.00	100.00
A19-PL-03	99.05	97.92	92.56	95.06	100.00	100.00	100.00	100.00
A19-PL-04	99.76	98.76	98.76	98.76	100.00	100.00	100.00	100.00
A19-PL-05	99.57	97.79	96.42	97.09	100.00	100.00	100.00	100.00
A19-PL-06	99.88	99.01	99.47	99.24	100.00	100.00	100.00	100.00
A22-PL-01	99.45	98.17	96.04	97.08	99.60	97.40	97.40	97.40
A22-PL-02	99.88	99.94	99.11	99.52	100.00	100.00	100.00	100.00
A22-PL-03	99.88	99.32	98.71	99.01	100.00	100.00	100.00	100.00
A22-PL-04	99.88	99.12	99.52	99.32	100.00	100.00	100.00	100.00
A22-PL-05	99.76	98.79	97.76	98.27	100.00	100.00	100.00	100.00
A25-PL-01	98.42	99.12	92.98	95.78	99.21	99.60	66.67	74.80
A25-PL-02	98.50	99.20	90.05	94.07	99.80	99.90	91.67	95.40
A25-PL-03	98.54	99.19	93.71	96.23	99.80	99.90	98.21	99.04
A25-PL-04	99.68	99.84	94.81	97.18	99.60	99.80	75.00	83.23
A25-PL-05	99.68	99.84	94.81	97.18	99.60	99.80	75.00	83.23
A25-PL-06	100.00	100.00	100.00	100.00	100.00	100.00	100.00	100.00
A29-PL-01	99.33	98.67	96.65	97.63	99.21	50.00	100.00	66.67
A29-PL-02	99.60	99.27	95.48	97.29	100.00	100.00	100.00	100.00
A29-PL-03	99.49	99.73	94.40	96.90	100.00	100.00	100.00	100.00
A29-PL-04	99.72	98.02	97.44	97.73	99.80	99.90	91.67	95.40
A29-PL-05	99.96	99.98	99.54	99.76	100.00	100.00	100.00	100.00
A35-PL-01	98.46	98.45	95.03	96.65	99.80	99.90	92.86	96.10
A35-PL-02	98.02	98.25	78.49	85.60	99.41	49.70	50.00	49.85

**Table 3 sensors-24-05858-t003:** Hyperparameter setting of methods.

Method	Hyperparameter
TC [43]	Batch size = 32, Learning rate = 0.0001, Epoch = 30
TAD [15]	Batch size = 16, Learning rate = 0.001, Epoch = 100, Num_heads = 4, hidden = 32
VAE [30]	$\mathrm{Batch}\;\mathrm{size}=32,\;\mathrm{Learning}\;\mathrm{rate}=0.001,\;\mathrm{Learning}\;\mathrm{rate}\;(\mathrm{Discriminator})=0.0002,\;\mathrm{Epoch}=30,\;\mathrm{filters}=32,\;\mathrm{Kernel}\;\mathrm{size}=7,\;\lambda_{V}$$=0.1,\;\lambda_{D}$= 0.5, N = 100

## Data Availability

Data utilized in this work are available from the corresponding author by request.

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
