# Peer review of "Self-Supervised Dam Deformation Anomaly Detection Based on Temporal–Spatial Contrast Learning"

_sensors, 2024, doi:10.3390/s24175858_

Round 1

Reviewer 1 Report

Comments and Suggestions for Authors

Anomalies could lead to dam deformation, which is critical in structure health monitor (SHM) and for failure warning. Facing the recent developments in AI models, the authors proposed an anomaly detection method employing a Spatio-temporal Contrastive Learning Pretraining (STCLP) to extract discriminative features from unlabeled datasets of dam deformation. The experimental results, whose data is from a Dam in China from 2012 to 2018, show that their method has better performance than other methods. Here are some major comments.

1.     There is only one way employed in weak augmentation and merely another one in strong augmentation, but however there are usually multiple approaches to augment the data in CS. Although it could be bit difficult in this case, as the signal is unidirectional time series, there should be other ways to enhance your data’s variety. By doing that, the model could possibly perform better.

2.     An interesting fact is that the authors claimed to use both time and space relationship for detection for the first time, however, the proposed model proceeds the two in serial. Have you ever considered input the data as an image, one axis denotes temporal, the other spatial (or different sensors)? Actually, this is a common approach in vibration/event detection employing Phi-OTDR.

3.     The authors have compared their methods with three other models, nevertheless, all three was proposed at least 2 years ago. For example, VAE was proposed in 1986 (even though your reference is the Li’s model by 2021), would it be more convincing if a more recent model is compared, for example, the GAN-VAE by Zhou et al. 2023 as you mentioned in the introduction? Btw, which TAD model have you employed for comparison (I suppose it would be Ref. 36)? It would be better to cite the origin in Table 3.

4.     Minor issues. The authors are missing in Ref. 32. The font of the last paragraph in the introduction is different from others. Fig. 9 is not clear with some curve covered by the legend.

Reviewer 2 Report

Comments and Suggestions for Authors

The manuscript presents a method for detecting anomalies in dam deformation data collected through Structural Health Monitoring devices. The proposed methodology is validated using real-world data from an arch dam equipped with pendulum systems for monitoring horizontal deformation. Additionally, the effectiveness of the method is demonstrated by comparing its results with those obtained from three established benchmark models.

The paper is interesting and properly organized. However, certain sections could be enhanced for clarity and precision. The authors are encouraged to consider the following suggestions before submitting a revised version:

Introduction:

The introduction should start with a brief overview of the importance of Structural Health Monitoring, similar to what is outlined in the abstract. This section should further expand on the topic with relevant references, emphasizing its crucial role in preserving structures and infrastructures, and most importantly, in safeguarding human lives. We recommend consulting the introductions of the following studies: https://doi.org/10.3390/app13063712 and 10.1088/1757-899X/603/5/052042.

Overall Text:

The language should be refined to improve readability and eliminate errors, such as the typo on line 494, where "blue squres denote" should be corrected.

References:

Please review and adjust the references to ensure they conform to the journal’s formatting guidelines.

Comments on the Quality of English Language

The english language should be refined to improve readability and eliminate typos
